# Valorisation of Spent Grain from Malt Whisky in the Spelt Pasta Formulation: Modelling and Optimization Study

Ancuța Chetrariu  and Adriana Dabija *

Faculty of Food Engineering, Stefan cel Mare University of Suceava, 720229 Suceava, Romania;
ancuta.chetrariu@fia.usv.ro
* Correspondence: adriana.dabija@fia.usv.ro; Tel.: +40-748-845-567

**Abstract:** Although durum wheat flour is conventionally used to produce pasta, in this study, emphasis was placed on the use of spelt flour in the formulation of the pasta recipe, with the replacement with spent grain obtained from distilleries for its content of fiber and protein. D-optimal design was used to optimize the influence of spent grain addition for the quality attributes of spelt pasta. In order to optimize the spelt pasta matrix, the spent grain content was varied (5%, 10%, 15%, and 20%) so that all responses were optimized (maximize cohesiveness, fracturability, proteins, total dietary fiber, total phenolic content, and antioxidant activity, minimize cooking loss, in-range firmness, and color paste). The optimal addition of spent grain in the spelt pasta recipe was 11.70%, yielding values with differences of less than 5% from the values predicted by the model and producing finished products with good nutritional properties without negative consequences on quality. Spent grain is a valuable byproduct that deserves to be used for fortification in order to obtain pro-health food. This study presents a formulation of spelt pasta with the addition of spent grain using mathematical modeling and statistical optimization.

**Keywords:** spelt pasta; optimization; spent grain; valorisation; pasta; pro-health food



## 1. Introduction

The agro-industry results in substantial quantities of by-products that have an increased content of organic compounds with significant environmental pollution impacts. On the other hand, the growing demand for food worldwide encourages the identification of alternatives at reasonable prices and nutritional qualities. Spent grain (SG) is the main by-product produced by the beer industry and the distillation industry after wort production, accounting for about 85% of the total by-products generated [1,2]. The whisky industry generates, on average, 8–15 L of effluent and 2.5–3.0 kg of spent grain for every liter of whisky produced, and brewing generates 0.2 kg of wet brewer's spent grain per liter of beer [2,3]. Spent grain is a valuable by-product rich in nutrients, such as dietary fibers (hemicellulose, cellulose, and lignin), digestible protein, monosaccharides (glucose, xylose, and arabinose), minerals, vitamins, and lipids, and the products enriched with it are called fortified foods [4–9]. Spent grain contains a substantial amount of bioactive compounds with high antioxidant capacities, such as hydroxycinnamic acids, especially ferulic and p-coumaric [10,11]. The composition of SG is affected by the differences in grain variety, harvest time and condition, the malting and mashing methods, and the type of adjunct grains, resulting in different variants of this by-product [12,13].

The capitalization of these by-products and the development of sustainable processes is an urgent necessity in the industry [12]. Spent grain is the insoluble fraction left after the malted barley is grounded and mixed with water to produce wort, i.e., the liquid fermentation medium [14]. Before using spent grain in foods, it needs to be dried and turned into flour, although its use has some limitations because of its brownish color and its flavor [3]. Dried spent grain has been studied and has been found to be applicable in

various food products with health-promoting properties, such as bread, cookies, flakes, pasta, biscuits, breakfast cereals, and snacks [15–19].

Spelt (*Triticum aestivum* var. *spelta*) has been cultivated extensively in Europe since ancient times and has returned to the attention of researchers and consumers because of its nutritional qualities and its resistance and ability to adapt to climatic conditions [9]. Consumer feedback shows that spelt flour-based products are more digestible and have therapeutic effects compared to products based on common wheat [14]. Climate change is another reason for the interest in spelt wheat since it has shown more resilience to drought and other extreme weather conditions than common wheat [8].

In the food industry, spelt grain has great technological potential for specialty breads, organic food, and food products, with characteristics that differ from regular wheat products [20]. Spelt grain can be used in brewing as an unmalted adjunct during mashing [21]. Other examples of foods that can be obtained from spelt grain are pasta, baked goods, high-fiber breakfast cereals, snacks, crackers, and beverages [6,22,23].

Spelt flour is used as a raw material to improve the nutritional and functional properties of pasta [24,25]. Spelt flour contains a higher fiber quantity and phenolic compounds, with health benefits such as reduced risk of some types of diseases; lower concentrations of blood lipids; stable blood glucose and insulin levels; a positive impact on minimizing fatigue and energy loss and removing toxins and lowering cholesterol levels in the blood; reduced risk of cardiovascular disease and better control of diabetes; the absence of constipation; and better weight management [14].

Consumers need good quality and affordable ready-to-eat food products that have a good glycemic index and long shelf life [26]. Among the most consumed foods that meet these requirements is pasta [2,27]. Furthermore, the production of pasta is simple (it is obtained by the extrusion of a mixture of flour from durum wheat and water), and it has a healthy nutrient content and good sensory qualities. Researchers have improved the matrix of pasta by enriching the nutritional potential by mixing durum wheat flour with different flours (e.g., pseudo-cereal flours and legume flours) [4,6,28]. Today, the use of high-functionality ingredients for value addition to pasta is a developing field of research to achieve innovative product categories, among which is the use of ancient grain flours, such as spelt flour [7]. Pasta is a good source of energy with high amounts of carbohydrates, moderate amounts of protein, and low lipids [10]. Fortifying these foods is a challenge, not only in terms of increasing nutritional properties and health but also the effects on cooking, texture, and sensory properties. The current research does not provide much information about the use of spelt flour and spent grain flour for pasta production, nutritional value, and spelt grain and spent grain's effects on health [6,7].

– The response surface method is used to find the optimal response and changes its direction because of the design variables, which can be seen as a visual graph. An experimental design should take into account the design constraints [29]. Some advantages of optimization methods are the following: computational efficiency; better description of the factors' influence in the process, both alone or in combinations; the relationship between the responses and the factors; and the achievement of a sustainable processing industry [30].

This study aimed to optimize the effect of the fortification of spent grain flour on the chemical composition, nutritional values, and selected quality properties of novel spelt pasta formulation. The novelty of this study is its use of spelt flour with the addition of spent grain from the process of producing whisky in the formulation of pasta recipes. Most previous studies have focused on spent grain from the beer industry [9,31–34], with few papers focusing on spent grain from distilleries; this is why little information is available about the nutritional value and fortification.

## 2. Materials and Methods

### 2.1. Materials

Spent grain from whisky production was collected from a local factory, Alexandrion Group (Ploiesti, Romania); wet spent grain was stored at $-18\ ^{\circ}$C, dried at 50 $^{\circ}$C for 24 h, ground by a mill, and then sieved. Spent grain flour was obtained from a fraction of less than 200 $\mu$m and was stored in paper bags at room temperature until further use. Spelt flour was purchased from a local market and had a Romanian origin.

All chemicals used in this paper were of analytical grade and were purchased from Sigma Aldrich (St. Louis, MO, USA).

### 2.2. Pasta Processing

The pasta dough was mixed with a Kitchen Aid mixer (Whirlpool Corporation, Benton Harbor, MI, USA) by adding spelt flour and different percentages of spent grain (5–20%) to obtain 32% dough moisture. The pasta was modeled after a resting time of 45 min, during which time the dough temperature was maintained at 40 $^{\circ}$C using a small macaroni mold. Pasta drying was performed in an air oven for 6 h at 40 $^{\circ}$C [26].

### 2.3. Dough Texture

The dough texture profile analysis (TPA) was performed using a Perten TVT-6700 texturometer (Perten Instruments, Hägersten, Sweden). From the dough, 50 g balls were prepared, which were analyzed to determine the firmness and cohesiveness. The sample was subjected to double compression at 50% height with a 35 mm cylinder probe at a speed of 5.0 mm/s and a trigger force of 20 g [35]. All measurements were made in triplicate.

### 2.4. Dry Pasta Color

A Konica Minolta CR-400 colorimeter (Tokyo, Japan) was used to measure the dry pasta color using CIELab color space coordinates, where L* values describe black to white (0 to 100), a* is the degree of redness (positive) or greenness (negative), and b* is yellowness (positive) or blueness (negative) [36]. The color change equation (Equation (1)) is described below:

$$\Delta E = \sqrt{(\Delta L)^2 + (\Delta a)^2 + (\Delta b)^2} \tag{1}$$

where $\Delta L$ = L* sample $-$ L* control, $\Delta a$ = a* sample $-$ a* control, and $\Delta b$ = b* sample $-$ b* control. All measurements were made in triplicate.

### 2.5. Dry Pasta Fracturability

A Perten TVT-6700 device (Perten Instruments, Hägersten, Sweden) equipped with an aluminum break rig set adjusted to a 10 mm width was used to determine dry pasta fracturability using the maximum force F (g) needed to break a pasta piece. The test speed was 2 mm/s and the trigger force was 5 g, and measurements were made in duplicate [37].

### 2.6. Crude Proteins and Total Dietary Fiber

Protein determination was carried out using Kjeldahl nitrogen analysis with a 5.7 conversion factor (Velp Scientifica, Usmate, Italy) [37]. A Megazyme total dietary fiber assay kit (Megazyme, Ireland) was used to determine the total dietary fiber using the enzymatic method [37].

### 2.7. Total Phenolic Content

A sample of 2 g of raw pasta was ground and mixed with 20 mL of methanol 80% (*v/v*) and sonicated for 40 min in a sonication bath at 37 $^{\circ}$C and 45 Hz; then, the mixture was centrifuged for 5 min at 4000 rpm [38]. Then, 0.2 mL of extract was mixed with 2 mL of Folin–Ciocalteu reagent, diluted to 1:10, and mixed with 1.8 mL of sodium carbonate 7.5% (*w/v*) in a tube. The mixture was left for 30 min at room temperature in the dark. The total polyphenolic content was determined at a 750 nm wavelength using a UV–VIS–

NIR Shimadzu 3600 (Shimadzu Corporation, Kyoto, Japan). The calibration curve of the polyphenols was performed by using gallic acid at concentrations of 10–200 mg/L with the regression coefficient $R^2$ = 0.99872 and the equation y = 0.00949x + 0.02950. The samples were analyzed in triplicate, and the results were expressed in µg gallic acid equivalents per gram (µg GAE/g).

### 2.8. Antioxidant Activity of Pasta

2,2-Diphenyl-1-picrylhydrazyl (DPPH) was used to assess the antioxidant activity. Following the methods of Iuga and Mironeasa [35], a sample of 2 g of raw pasta was ground and mixed with 20 mL of methanol 80% (*v*/*v*) in a sonication bath at 37 °C and 45 Hz for 40 min and then centrifuged for 5 min at 4000 rpm. Then, 2 mL of extract was mixed with 2 mL of DPPH solution 0.1 mM in methanol [30]. Absorbance was determined at 517 nm using a UV–VIS–NIR spectrophotometer (Shimadzu Corporation, Kyoto, Japan) against a blank sample. The antioxidant capacity was measured in triplicate using Equation (2).

$$\% \text{ Inhibition of DPPH} = [(1 - \text{As}/\text{Ab})] \times 100 \qquad (2)$$

where As = absorbance of the sample and Ab = absorbance of the blank sample.

### 2.9. Pasta Cooking Behavior

Cooking loss (CL) was determined gravimetrically by the evaporation of the water that resulted after boiling for an optimal cooking time of 10 g of pasta in 100 mL of water without salt addition. The residue was weighed and expressed as grams of matter loss per 100 g of pasta [34]. Cooking loss was determined in triplicate.

### 2.10. Optimization of Spent Grain Level and Model Validation

A trial version of Design-Expert software (Stat-Ease, Inc., Minneapolis, MN, USA) was used for experimental design development, data analysis, regression modeling, and optimization. To validate the model, pasta produced with the optimal level of spent grain was analyzed, and real values were compared to the control sample made with no spent grain addition. D-optimal design with one factor varied at four levels (5%, 10%, 15%, and 20%) was used for the assessment of the spent grain addition on spelt pasta quality. The model's fit was evaluated through a sequential Fisher test, coefficients of determination ($R^2$), and adjusted coefficient of determination (Adj.-$R^2$) at a 95% confidence level.

### 2.11. Sensory Analysis of Pasta

For organoleptic analysis, samples of pasta cooked at a ratio of 1:10 (pasta:water) and served without any accompaniment were given to a group of seven experts selected and trained in order to identify particular attributes and in terms of sensory vocabulary. In this respect, the preferential method with a nine-point evaluation scale was used. Among the parameters evaluated were appearance, color, smell, taste, firmness during chewing, and general acceptability. During sample evaluation, talking among panelists was not allowed.

### 2.12. Statistical Analysis

All measurements in the present study were made in triplicate. Results are presented as means ± standard deviation (SD). XLSTAT for Excel 2021 (Addinsoft, New York, NY, USA) was used for the statistical analysis of the data. To evaluate significant differences ($p < 0.05$) among samples, the *t*-test was used.

## 3. Results and Discussion

### 3.1. Model Fitting and Statistical Analysis

A D-optimal design was used, which has an independent variable with four levels and three replications at the center point. The D-optimal design was used for the optimization of the formulation of spelt pasta, and different parameters of the optimized formulation

were evaluated. The linear and quadratic models that were fitted on each response were evaluated. The chemical composition of spelt flour is described in Table 1, which we obtained from a previous study [39].

**Table 1.** Chemical composition of spent grain flour and spelt flour [39].

| Chemical Composition | Spent Grain Flour | Spelt Flour |
| --- | --- | --- |
| lipids | $7.11 \pm 0.39$ | $3 \pm 0.01$ |
| fiber | $22.67 \pm 0.42$ | $8 \pm 0.05$ |
| protein | $18.88 \pm 0.37$ | $14 \pm 0.09$ |
| ash | $3.47 \pm 0.02$ | $2.11 \pm 0.04$ |
| moisture | $5.04 \pm 0.42$ | $11.26 \pm 0.08$ |

Results are expressed as g reported at 100 g dry matter.

The mathematical models were significant and presented the responses accurately; in all cases, the F-value was significant ($p < 0.01$), and $R^2$ values were more than 0.87, according to the ANOVA results presented in Table 2.

**Table 2.** ANOVA results for spent grain pasta and dough.

| Response | Model | F-Value | *p*-Value | $R^2$ | Adj.-$R^2$ |
| --- | --- | --- | --- | --- | --- |
| Cohesiveness | quadratic | 14.6 | <0.01 | 0.87 | 0.81 |
| Firmness | linear | 100.76 | <0.01 | 0.95 | 0.94 |
| Color | linear | 15.18 | <0.01 | 0.75 | 0.70 |
| Fracturability | quadratic | 105.81 | <0.01 | 0.98 | 0.97 |
| Crude proteins | quadratic | 415.93 | <0.01 | 0.99 | 0.99 |
| Total dietary fiber | linear | 247.60 | <0.01 | 0.98 | 0.97 |
| Total phenolic content | linear | 364.43 | <0.01 | 0.98 | 0.98 |
| Antioxidant activity | quadratic | 63.25 | <0.01 | 0.96 | 0.95 |
| Cooking loss | quadratic | 253.08 | <0.01 | 0.99 | 0.98 |

$R^2$ represents the coefficient of determination, defining the ratio of the variation of the answers that is explained by the model. The closer the $R^2$ is to 1, the better it fits the model. Adjusted $R^2$ is a correction of the value $R^2$ dependent on the number of degrees of freedom [40]. The color results, firmness, total dietary fiber, and total polyphenol content were fitted to the linear model with 75%, 95%, 98%, and 98% of the data variation explained, respectively. The quadratic model explained 87%, 98%, 99%, and 99%, respectively, of the data variation for dough cohesivity, fracturability, crude proteins, and cooking loss.

### 3.2. Optimization of Parameters and Validation of the Models

Small particle size (200 μm in this study) did not give significant differences in the cohesiveness, with the linear term having the most significant influence in Equation (3) ($p < 0.01$). Multiple regression analysis was used to analyze the experimental data, and thus a quadratic polynomial equation was obtained as follows (*** $p < 0.001$; ** $p < 0.01$; * $p < 0.05$). A small decrease in cohesiveness with spent grain addition level increase was observed in Figure 1A. Cohesiveness is the ability of the material to withstand two successive compressions [41]. Petitot et al. (2010) showed in their study that the addition of flour to pasta with vegetables had no impact on cohesiveness [41].

$$\text{Cohesiveness} = 0.289 - 0.021A^{***} + 0.008A^2 \qquad (3)$$

Firmness represents the cutting force required to penetrate pasta dough and was significantly increased by the linear term in Equation (4). The shaping of the pasta is influenced by the firmness of the dough, with a stronger dough desirable for shaping the short pasta to keep its shape [42,43]. A rise in firmness with an increase in spent grain addition level was observed (Figure 1B). Zhao et al. reported that a pasta enriched with legume flour (navy bean, pinto bean, lentil, and green pea) or protein concentrates caused

an increase in pasta firmness [44]. Tazrart et al. showed that an increase in bean flour levels resulted in a slight increase in pasta dough firmness [45].

$$\text{Firmness} = 5861.87 + 533.081A^{***} \tag{4}$$

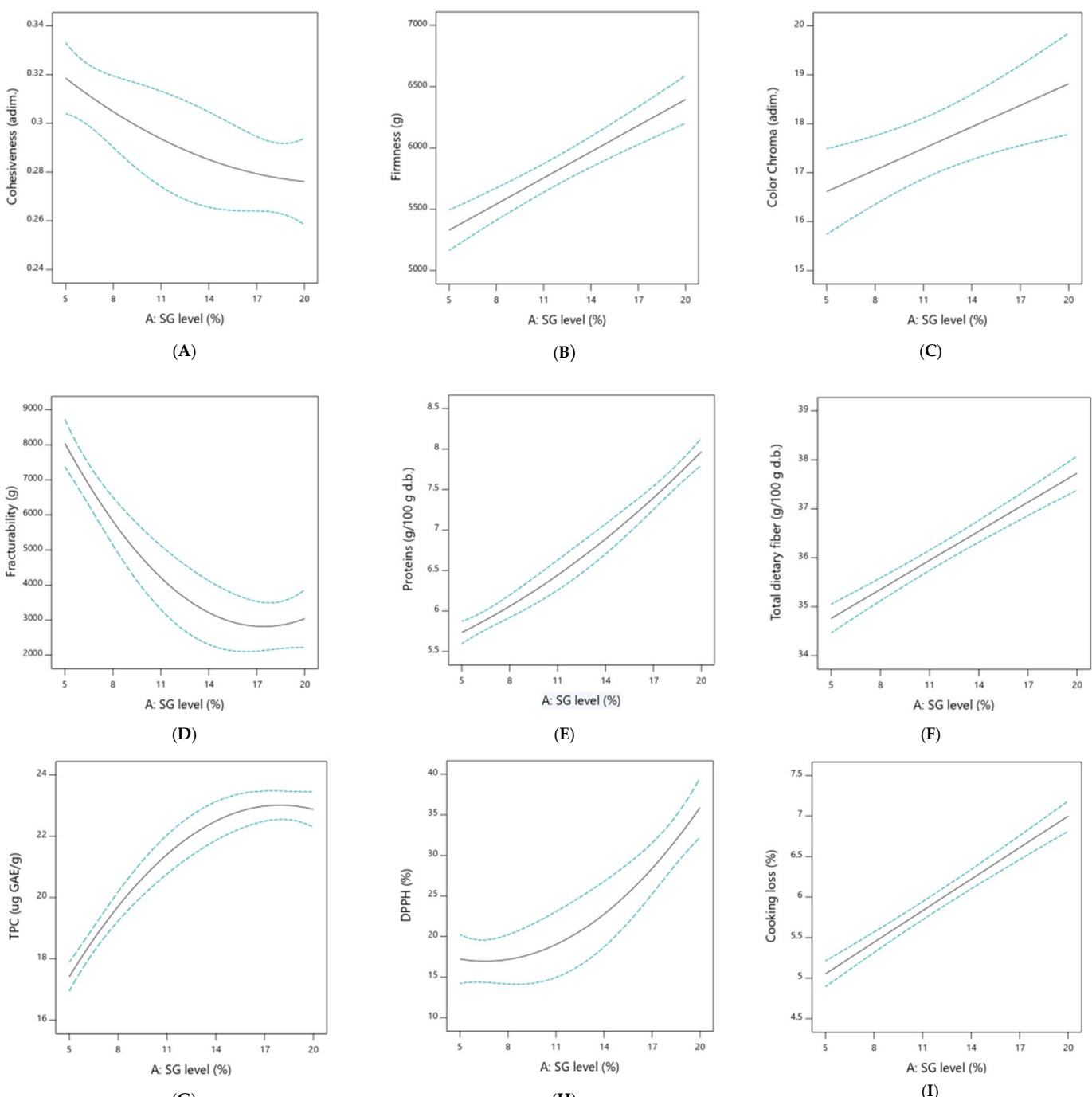

**Figure 1.** Effect of spent grain on (**A**) dough cohesiveness, (**B**) firmness of pasta dough, (**C**) pasta color, (**D**) pasta fracturability, (**E**) protein content, (**F**) total dietary fiber content, (**G**) total phenolic content, (**H**) antioxidant activity of pasta, and (**I**) pasta cooking loss.

The first factor governing food acceptability is the product appearance; in the case of spent grain incorporation, the color of the products is darker with increased incorporation. The same results were observed in other studies [34,46–48]. Products with high fiber

contents have a positive feature of dark coloring, which consumers associate with products rich in fiber [34]. Pasta color was affected by spent grain incorporation; a significant increase in C* was observed with the additional level increase. The linear term had the greatest significant influence, as can be seen in Equation (5). Figure 1C shows the increase of color with spent grain addition.

$$\text{Color (C*)} = 17.715 + 1.100A^{***} \tag{5}$$

Fracturability (g) is the maximum force needed to break a pasta piece [35,37]. The fracturability of dried pasta decreases with the increase of the addition of spent grain; the linear term had the greatest negative effect on the response, while the quadratic term had a significant positive effect on the dry pasta fracturability (Equation (6)). The decrease of fracturability with spent grain addition is shown in Figure 1D; the enriched fish pasta had the same decrease of fracturability, possibly because of the fiber content or fat composition of each assortment of the pasta [49]. Naibaho and Korzeniowska showed that the addition of spent grain in corn snacks decreased the fracturability of the products [15].

$$\text{Fracturability} = 3635.83 - 2505.19A^{***} + 1909.53A^{2***} \tag{6}$$

A protein content increase with spent grain addition can be observed in Figure 1E. Pasta proteins were significantly influenced by the linear and quadratic terms in Equation (7). This tendency for protein levels to increase with increased levels of added spent grain was also observed in other studies [45,50].

$$\text{Proteins} = 6.658 + 1.115A^{***} + 0.192A^{2**} \tag{7}$$

A total dietary fiber increase with spent grain addition can be observed in Figure 1F. The total dietary fiber of pasta was significantly influenced by the linear term in Equation (8). Nocente et al. obtained an increased total dietary fiber content with the increase of spent grain in pasta production [32]. The same results were obtained in other studies [14,31,45,46,50,51].

$$\text{Total dietary fiber} = 36.244 + 1.483A^{***} \tag{8}$$

The total phenolic content increased with spent grain addition (Figure 1G), and from Equation (9), it can be observed that the linear term had the highest positive effect and the quadratic term had a significant negative effect on the response. Spelt pasta can be used to produce precooked pasta with a high content of biologically active compounds compared to refined flour [14]. Pasta supplemented with native spent grain or fermented spent grain had an increase in phenolic content [34]. Spinelli et al. obtained fortified spent grain pasta with an increase in phenolic content [33].

$$\text{Total phenolic content} = 22.023 + 2.728A^{***} - 1.875A^{2***} \tag{9}$$

An antioxidant activity increase with spent grain addition can be observed in Figure 1H. The DPPH test was significantly influenced by the linear and quadratic terms in Equation (10). In their study, Reis and Abu-Ghannam obtained snacks with 10–40% spent grain addition, which increased the phenolic content and antioxidant activity [52]. Nocente et al. obtained spent grain pasta with a total antioxidant capacity of up to 19% compared with the control durum flour pasta [48].

$$\text{Antioxidant activity} = 20.654 + 9.327A^{***} + 5.901A^{2***} \tag{10}$$

A value below 12% of cooking loss indicates good quality products, and cooking loss is considered an indicator of the general cooking performance of pasta [53–55]. Cooking loss increased with the increase of spent grain addition (Figure 1I), but cooking losses were

below the values reported for good quality durum wheat pasta (<7%) [56]. The linear term had the most significant positive effect, as can be seen in Equation (11). Cooking loss increases as cohesiveness decreases, according to Vital et al. [53]. Pasta rich in fiber tends to have a higher cooking loss, probably due to the weakening of the protein network by the presence of fiber content [32]. According to Rousta et al., as the amount of protein and fiber increased in the pasta recipe, the cooking loss increased [27].

$$\text{Cooking loss} = 6.026 + 0.972A^{***} \qquad (11)$$

The predicted results for the responses were verified, and differences of ≤5% were obtained between the predicted and verified values.

### 3.3. Optimization of Spent Grain Level and Model Validation

Optimizing the spent grain level in the pasta recipe showed that spent grain can be added at a rate of 11.70% with desirability of 0.371 without affecting the quality characteristics, obtaining maximum nutritional benefits, applying the constraints (Figure 2).

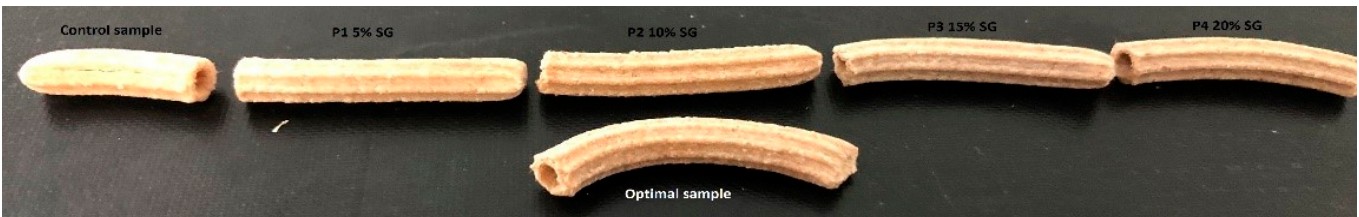

**Figure 2.** Pasta samples obtained in this research study.

The results obtained made it possible to locate the lower and upper limits according to the target, as shown in Table 3. The following objectives were chosen to take into account the measured quality characteristics: minimization of cooking losses; maximization of cohesiveness, fracturability, crude proteins, and total dietary fiber content; and total phenolic content and antioxidant activity, keeping in range the firmness and color of pasta.

**Table 3.** Optimization constraints for pasta recipe.

| Name | Goal | Lower Limit | Upper Limit | Importance |
|---|---|---|---|---|
| A:SG level (%) | in range | 5 | 20 | 3 |
| Cohesiveness (adim.) | maximize | 0.27 | 0.329 | 3 |
| Firmness (g) | in range | 5359 | 6437 | 3 |
| Color chroma (adim.) | in range | 15.53 | 18.88 | 3 |
| Fracturability (g) | maximize | 2935 | 8120 | 3 |
| Crude proteins (g/100 g d.b.) | maximize | 5.68 | 7.99 | 3 |
| Total dietary fiber (g/100 g d.b.) | maximize | 34.64 | 37.95 | 3 |
| Total phenolic content (µg GAE/g) | maximize | 17.21 | 23.03 | 3 |
| Antioxidant activity (% inhibition) | maximize | 15.6 | 35.55 | 3 |
| Cooking loss (%) | minimize | 5.01 | 6.992 | 3 |

A sample was made with the optimal value predicted to validate the model, and the answers were evaluated in triplicate; the predicted value and verified value of the optimal sample can be seen in Table 4.

A pasta sample was obtained with the optimal level of spent grain resulting after the optimization process. All responses were checked in triplicate, and the values of the experimental results were less than 5% different from the predicted ones. The addition of spent grain in the pasta recipe in the presented conditions allowed the production of pasta with a short preparation time and an attractive texture after hydration with warm water.

**Table 4.** Predicted and verified optimal sample.

| Factor | Spent Grain Pasta | | |
|---|---|---|---|
| | **Predicted Value** | **Verified Value** | **Relative Deviation * (%)** |
| SG level | 11.70 | 11.70 | |
| Cohesiveness (adim.) | 0.291 [a] | 0.302 [a] | 3.64 |
| Firmness (g) | 5805.01 [a] | 5786.06 [a] | −0.33 |
| Color (adim.) | 17.59 [a] | 17.69 [a] | 0.52 |
| Fracturability (g) | 3924.74 [a] | 3976.25 [a] | 1.30 |
| Crude proteins (g/100 g d.b.) | 6.54 [a] | 6.67 [b] | 1.96 |
| Total dietary fiber (g/100 g d.b.) | 36.08 [a] | 36.42 [a] | 0.92 |
| Total phenolic content (µg GAE/g) | 21.71 [a] | 22.84 [a] | 4.94 |
| Antioxidant activity (% inhibition) | 19.72 [a] | 20.12 [a] | 1.96 |
| Cooking loss (%) | 5.92 [a] | 5.71 [a] | −3.71 |

* relative deviation = ((experimental value − predicted value)/experimental value) × 100, [a,b] indicates that differences among predicted and observed values are significantly different ($p < 0.05$).

### 3.4. Sensory Analysis of Pasta

The appearance of the pasta with spent grain addition was pleasant, obtaining a slightly higher score than the control sample obtained only from spelt flour. The panelists also appreciated the color and smell parameters positively compared to the reference sample. The sensory properties of cooked pasta are shown in Figure 3. The highest values were obtained for color, appearance, and smell parameters, while the taste parameter obtained the lowest score.

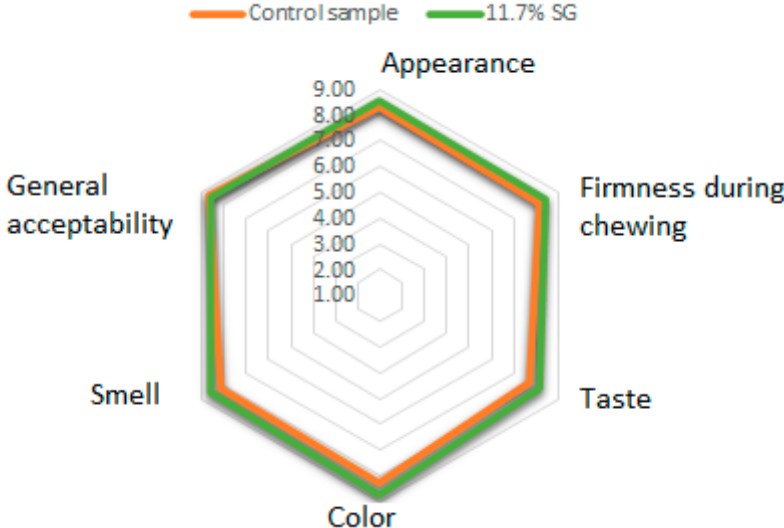

**Figure 3.** Sensory scores for pasta.

### 4. Conclusions

The purpose of this work was to establish the optimal value of spent grain in spelt pasta with the preservation in parameters of the desired responses. Spent grain is an important source of protein, fiber, and compounds with antioxidant properties. As the results show, the optimal value for spelt pasta formulation with a good balance between sensory and nutritional aspects was 11.70%. The optimal level of spent grain was used without compromising the acceptability of the product. Spelt pasta fortified with spent grain is included in products with a high fiber and protein content and with antioxidant activity and high polyphenol content. The color of the pasta obtained was acceptable, and the cooking losses were within the limit of 12%, which fits them into good quality products. These results show that spent grain can be used successfully in the recipe for fortified pasta, obtaining high-quality products. Spent grain flours can be used in food formulations

because of their potential to improve the nutritional quality of the product and may have a lower glycemic index compared to pasta based on white flour from durum wheat. The valorization of spent grain can improve the sustainability of the brewing process and the whisky production process. With the help of experimental modeling, pasta recipes can be developed to meet consumer requirements or by directing recipes to certain categories of consumers, depending on needs.

**Author Contributions:** Conceptualization, A.C. and A.D.; Methodology, A.C.; Formal Analysis, A.C. and A.D.; Investigation, A.C.; Resources, A.D.; Writing—Original Draft Preparation, A.C.; Writing—Review and Editing, A.C. and A.D. All authors have read and agreed to the published version of the manuscript.

**Funding:** No funding was obtained for this work.

**Institutional Review Board Statement:** Not applicable.

**Informed Consent Statement:** Not applicable.

**Data Availability Statement:** The data presented in this study are available in this article.

**Acknowledgments:** We would like to thank Alexandrion Group Romania for providing the spent grain used in this work. The authors acknowledge financial support from Stefan cel Mare University of Suceava, Romania.

**Conflicts of Interest:** The authors declare no conflict of interest.

**Sample Availability:** Samples were available from Alexandrion Group Romania.

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
