# Peer review of "Valorisation of Spent Grain from Malt Whisky in the Spelt Pasta Formulation: Modelling and Optimization Study"

_applsci, doi:10.3390/app12031441_

Round 1
Reviewer 1 Report
The manuscript ‘Valorisation of Spent Grain from Malt Whisky in the Spelt Pasta Formulation. Modelling and Optimization Study’ is focused on the value addition of the spent grain in spelt pasta formulation, which has industrial significance. The authors have attempted to optimize pasta formulations fortified with spent grain using mathematical modelling which is appreciable. The manuscript is well designed, well written and well presented. However, the authors may consider incorporating few in the manuscript prior to acceptance.
- Title: Valorisation of Spent Grain from Malt Whisky in the Spelt Pasta Formulation. Modelling and Optimization Study. Author may use colon ‘:’ to make a single sentenced title instead of ‘.’.
- Line 18-20: In this study, it was done the optimization…..Pl re-write the sentence. Please avoid long sentences and use short and clear sentences throughout the text.
- Line 24-27: Please make short sentences for smooth reading and understanding… The sentence may be revised as “The agro-industries resulted in substantial quantities of by-products that have an increased content of organic compounds with significant environmental pollution impacts. On the other hand, the growing demand for food worldwide encourages the finding of alternatives at reasonable prices and nutritional qualities.
- Line 41-42: May be revised as “The capitalization of these by-products and the development of sustainable processes is an urgent necessity in the industry [12]”.
- Line 49-51: ‘at the moment he knows a renewed interest…’. Unclear statement. Please revise the sentence.
- Line 58-63: Please delete some type of diseases ‘…risk of breast cancer………’
- Line 63-65: Unclear statement. Please re-write directly that ‘the consumers need good quality and affordable ready-to-eat food products having good glycemic index and long shelf life’. Pl modify accordingly.
- Line 67: ‘Furthermore, the production of pasta it’s facile, it is obtained’. Please rectify the sentence.
- Line 79: …relatively little information is available on the nutritional value…
- Line 81: ‘Response Surface Method’… Please introduce properly. I feel line 81-92 is undesired may be deleted. You may write the information in a single sentence with a coherence to the previous paragraph.
- Line 93: ‘addition’ may be replaced by ‘fortification’
- Line 95-99: Please delete “in order to………….consumers’. The novelty of the work well understood. I think it is not required to repeat.
- Lime 99-100: May be covered under relatively little information is available on fortification and nutritive value [Line 79].
- Line 103: ‘kindly provided’ may be replaced with ‘were collected’
- Line 104: Wet spent grain….make it another sentence.
- Section 2.2 Pasta Processing: Please add a reference if followed a published work.
- Line 147: Please check the original reference of AOAC 2011.25 [B.V. McCleary, J.W. DeVries, J.I. Rader, G. Cohen, L. Prosky, D.C. Mugford, et al. Determination of insoluble, soluble, and total dietary fiber (CODEX definition) by enzymatic-gravimetric method and liquid chromatography: Collaborative study Journal of AOAC International, 95 (3) (2012), pp. 824-844
- Section sub headings 2.1, 2.2, 2.3….3.1, 3.2, 3.3 etc., please follow the same pattern ‘Sentence case’ or ‘Title Case’ as per the authors guideline.
- The results of the study were well presented with discussion; however, the ‘Discussion’ heading is missing. Please see ‘Instruction to author’, and make the heading 3. Results, 4. Discussion or 3. Results and Discussion. Additionally, the author may interpret the result of the nutritional content in spent grain fortified spelt pasta at different concentration along with the linear regression results.
- The figures 1-9 may be arranged in a panel of Fig. 1A-I. Effect of spent grain on A. dough cohesiveness, B. firmness of pasta dough, C. pasta color….and so on. Legends on X- and Y-axes may be increased to a visible size.
- 10 may be represented as Fig. 2 with the visible lebels.
- Line 333: good balance between sensory and nutritional aspects was 11.70%. Have you performed any sensory test? Please mention the method followed and the size of the panel for the sensory test.
- Line 336: Agreeable or acceptable.
The manuscript may be accepted after minor revision.
Good luck.

Author Response
Dear Referee,
We would like to thank the referee for the close reading and for the proper suggestions.
We hope that we provide all the answers to the reviewer’s comments.
Thank you very much for the recommendations to publish our paper entitled “Valorisation of Spent Grain from Malt Whisky in the Spelt Pasta Formulation: Modelling and Optimization Study”.
The present version of the paper has been revised according to the reviewer’s suggestions.
We uploaded the corrected version of the article for which we used the Track changes function.
The manuscript ‘Valorisation of Spent Grain from Malt Whisky in the Spelt Pasta Formulation. Modelling and Optimization Study’ is focused on the value addition of the spent grain in spelt pasta formulation, which has industrial significance. The authors have attempted to optimize pasta formulations fortified with spent grain using mathematical modelling which is appreciable. The manuscript is well designed, well written and well presented. However, the authors may consider incorporating few in the manuscript prior to acceptance.
- Title: Valorisation of Spent Grain from Malt Whisky in the Spelt Pasta Formulation. Modelling and Optimization Study. Author may use colon ‘:’ to make a single sentenced title instead of ‘.’.
We would like to thank to the referee for her/his remarks. We made the changes according to the referee suggestion.
- Line 18-20: In this study, it was done the optimization…..Pl re-write the sentence. Please avoid long sentences and use short and clear sentences throughout the text. This study presents a formulation of spelt pasta with the addition of spent grain using the mathematical modeling and statistical optimization
We would like to thank to the referee for her/his remarks. We made the changes according to the referee suggestions.
- Line 24-27: Please make short sentences for smooth reading and understanding… The sentence may be revised as “The agro-industries resulted in substantial quantities of by-products that have an increased content of organic compounds with significant environmental pollution impacts. On the other hand, the growing demand for food worldwide encourages the finding of alternatives at reasonable prices and nutritional qualities.
We made the changes according to the referee suggestions.
- Line 41-42: May be revised as “The capitalization of these by-products and the development of sustainable processes is an urgent necessity in the industry [12]”.
Thank you for your suggestions.
- Line 49-51: ‘at the moment he knows a renewed interest…’. Unclear statement. Please revise the sentence.
We revised the sentence.
- Line 58-63: Please delete some type of diseases ‘…risk of breast cancer………’
We deleted that information.
- Line 63-65: Unclear statement. Please re-write directly that ‘the consumers need good quality and affordable ready-to-eat food products having good glycemic index and long shelf life’. Pl modify accordingly.
Thank you for your suggestion.
- Line 67: ‘Furthermore, the production of pasta it’s facile, it is obtained’. Please rectify the sentence.
We rectified that sentence.
- Line 79: …relatively little information is available on the nutritional value…
We rectified that paragraph.
- Line 81: ‘Response Surface Method’… Please introduce properly. I feel line 81-92 is undesired may be deleted. You may write the information in a single sentence with a coherence to the previous paragraph.
We rectified that paragraph.
- Line 93: ‘addition’ may be replaced by ‘fortification’
Thank you for your suggestion.
- Line 95-99: Please delete “in order to………….consumers’. The novelty of the work well understood. I think it is not required to repeat.
We deleted that information.
- Lime 99-100: May be covered under relatively little information is available on fortification and nutritive value [Line 79].
Thank you for your suggestion.
- Line 103: ‘kindly provided’ may be replaced with ‘were collected’
We made the changes required.
- Line 104: Wet spent grain….make it another sentence.
We rectified that sentence.
- Section 2.2 Pasta Processing: Please add a reference if followed a published work.
Thank you for your suggestion. We made the changes required.
- Line 147: Please check the original reference of AOAC 2011.25 [B.V. McCleary, J.W. DeVries, J.I. Rader, G. Cohen, L. Prosky, D.C. Mugford, et al. Determination of insoluble, soluble, and total dietary fiber (CODEX definition) by enzymatic-gravimetric method and liquid chromatography: Collaborative study Journal of AOAC International, 95 (3) (2012), pp. 824-844
Thank you for your suggestion.
- Section sub headings 2.1, 2.2, 2.3….3.1, 3.2, 3.3 etc., please follow the same pattern ‘Sentence case’ or ‘Title Case’ as per the authors guideline.
Thank you for your suggestion.
- The results of the study were well presented with discussion; however, the ‘Discussion’ heading is missing. Please see ‘Instruction to author’, and make the heading 3. Results, 4. Discussion or 3. Results and Discussion. Additionally, the author may interpret the result of the nutritional content in spent grain fortified spelt pasta at different concentration along with the linear regression results.
Thank you for your suggestion. We made the change.
- The figures 1-9 may be arranged in a panel of Fig. 1A-I. Effect of spent grain on A. dough cohesiveness, B. firmness of pasta dough, C. pasta color….and so on. Legends on X- and Y-axes may be increased to a visible size.
Thank you for your suggestion. We made the changes.
- 10 may be represented as Fig. 2 with the visible lebels.
Thank you for your suggestion. We made the change.
- Line 333: good balance between sensory and nutritional aspects was 11.70%. Have you performed any sensory test? Please mention the method followed and the size of the panel for the sensory test.
Thank you for your suggestion. We made a sensory analysis and added a new paragraph in the manuscript.
- Line 336: Agreeable or acceptable.
Thank you for your suggestion.
The manuscript may be accepted after minor revision.
Good luck.

Reviewer 2 Report
The presented manuscript is a valuable study, followed by careful statistical analysis of the research results. The results of chemical tests and measurements of the texture parameters of the raw product are also valuable.
However, in my opinion, to complete the fully explain of this issue, i.e. recipe optimization, it is advisable to supplement the determinations in the field of cooking quality of pasta (optimal cooking time, weight/volume increase index) and sensory assessment.
Did the authors analyze the texture of the pasta after cooking? These results can be combined with the sensory evaluation of the cooked pasta. As are the results of the color measurements or ΔE calculations.
I also submit a question whether this work is part 1 and the authors plan to submit part 2, in which optimization of pasta recipe with the use of spent grain also in the study of texture, color, sensory evaluation and nutritional value of pasta after cooking. In a way, such a suggestion is contained in the last sentence of the conclusions.

Author Response
Dear Referee,
We would like to thank the referee for the close reading and for the proper suggestions.
We hope that we provide all the answers to the reviewer’s comments.
Thank you very much for the recommendations to publish our paper entitled “Valorisation of Spent Grain from Malt Whisky in the Spelt Pasta Formulation: Modelling and Optimization Study”.
The present version of the paper has been revised according to the reviewer’s suggestions.
We uploaded the corrected version of the article for which we used the Track changes function.
The presented manuscript is a valuable study, followed by careful statistical analysis of the research results. The results of chemical tests and measurements of the texture parameters of the raw product are also valuable.
- However, in my opinion, to complete the fully explain of this issue, i.e. recipe optimization, it is advisable to supplement the determinations in the field of cooking quality of pasta (optimal cooking time, weight/volume increase index) and sensory assessment.
Thank you for your suggestion. We made a sensory analysis and added a new paragraph in the manuscript.
- Did the authors analyze the texture of the pasta after cooking? These results can be combined with the sensory evaluation of the cooked pasta. As are the results of the color measurements or ΔE calculations.
Thank you for your suggestion. We were working on that.
- I also submit a question whether this work is part 1 and the authors plan to submit part 2, in which optimization of pasta recipe with the use of spent grain also in the study of texture, color, sensory evaluation and nutritional value of pasta after cooking. In a way, such a suggestion is contained in the last sentence of the conclusions.
Thank you for the pertinent observation, indeed we are working on continuing and completing the researches that will materialize in some future studies.

Reviewer 3 Report
Detailed recommendation:
Abstract: in my opinion you should add more data to the abstract section.
Key words: please add: pasta, pro-health food
Introduction: give more information about application possibility of spelt grain in food industry.
What was the dry mass of pasta after drying?
Give more information about prepare sample to the textural analysis.
In which replication was made color analysis?
Why Dry Pasta Fracturability was made in duplicate? In research triplicate is obligatory.
Has an organoleptic evaluation been carried out? In my opinion this is obligatory.
Please consider in conclusion whether such pasta may have a lower glycemic index compared to pasta based on white flour from durum wheat.
Author Response
Dear Referee,
We would like to thank the referee for the close reading and for the proper suggestions.
We hope that we provide all the answers to the reviewer’s comments.
Thank you very much for the recommendations to publish our paper entitled “Valorisation of Spent Grain from Malt Whisky in the Spelt Pasta Formulation: Modelling and Optimization Study”.
The present version of the paper has been revised according to the reviewer’s suggestions.
We uploaded the corrected version of the article for which we used the Track changes function.
Detailed recommendation:
- Abstract: in my opinion you should add more data to the abstract section.
Thank you for your suggestion. We made some changes.
- Key words: please add: pasta, pro-health food
Thank you for your suggestion.
- Introduction: give more information about application possibility of spelt grain in food industry.
Thank you for your suggestions. We added more information about application possibility of spelt grain in food industry.
- What was the dry mass of pasta after drying?
Thank you for the pertinent observation, in this study we did not make a ratio between the meal of pasta after drying compared to the raw pasta, but we will consider in future research.
- Give more information about prepare sample to the textural analysis.
Thank you for your suggestion.
- In which replication was made color analysis?
The color analysis was made in triplicate.
- Why Dry Pasta Fracturability was made in duplicate? In research triplicate is obligatory.
Thank you for your suggestion.
- Has an organoleptic evaluation been carried out? In my opinion this is obligatory.
Yes, we made a sensory analysis, we added new paragraph in the manuscript.
- Please consider in conclusion whether such pasta may have a lower glycemic index compared to pasta based on white flour from durum wheat.
Thank you for your suggestion.

Round 2
Reviewer 1 Report
The manuscript ‘Valorisation of Spent Grain from Malt Whisky in the Spelt Pasta Formulation. Modelling and Optimization Study’ has been revised substantially. I have a very few observations which may be addressed while the manuscript be processed with production.
- Please reduce the space in line no. 19…’ ‘ This study presents..
- 83..Please start the sentence without‘-‘ Response surface method..
- Section sub headings 2.1, 2.2, 2.3….3.1, 3.2, 3.3 etc., please follow the same pattern ‘Sentence case’ or ‘Title Case’ as per the authors guideline.
2.3. Dough texture
2.4. Dry pasta color
2.5. Dry Pasta Fracturability
2.8. Antioxidant Activity of pasta
Similarly, pl check 3.1, 3.2, 3.3 etc.
- In Figure 1, please mention A, B, C, D…. in one corner inside the graph. Please increase the graph area and reduce the gaps between the graph. Please make it a close panel, it may look good.
- In Figure 2, the black labels on black background may be turned into white. Font of the label may be increased to a visible size.
- Why 3 title is italics…please check.
The manuscript may be accepted for publication.
Cheers!!!

Reviewer 2 Report
Dear Authors,
thank you for preparing the answer to my review of the article. I note that some of my comments have been applied to the revised manuscript. I assume that the aim of the authors of this manuscript was the Modelling and Optimization Study. I also accept the answer regarding the confirmation of the continuation of research in the field of the study discussed in this article, which will be published in subsequent articles supplementing and completing the discussion of the entire issue of the quality of wheat pasta with the addition of spent grain, including culinary features/cooking quality of pasta, sensory evaluation, texture parameters and nutritional value of the cooked pasta.
Reviewer 3 Report
Dear Authors
Thank you for your answer. The Authors' responses are satisfactory and the manuscript has a higher level.